# Optimal Design of Relay Coil Inductance to Improve Transmission Efficiency of Four-Coil Magnetic Resonance Wireless Power Transmission Systems

Min-Wook Hwang, Young-Min Kwon and Kwang-Cheol Ko *

Department of Electrical Engineering, Hanyang University, 222, Wangsimni-ro, Seongdong-gu, Seoul 04763, Republic of Korea; h_minwook@naver.com (M.-W.H.); dyeco@hanmail.net (Y.-M.K.)
* Correspondence: kwang@hanyang.ac.kr

**Abstract:** Magnetic resonance wireless power transmission consists of a source coil and relay coil (transmission coil (Tx-coil), receiving coil (Rx-coil)). The relay coil is designed with windings and a series capacitor, which are resonant with the input voltage frequency. Magnetic resonant wireless power transmission by a relay coil enables the transmission of power from a few centimeters to several meters. Recently, research has been conducted on the shape and material of each coil to increase the transmission distance. However, limitations remain with respect to increasing the transmission distance. Specifically, the optimization of the electrical characteristics of the relay coil is necessary to increase the transmission distance and improve efficiency. In this study, we configured the inductance of the relay coil to be approximately 95 µH, 270 µH, and 630 µH. Accordingly, we designed the series capacitors to have the same resonant frequency and analyzed the transmission characteristics of each relay coil. We confirmed that as the inductance increased, the transmission efficiency increased by up to 10%. The relay coil was designed to have an inductance of approximately three to six times that of the source coil (load coil). Thus, the optimal design of the relay coil is believed to be the most efficient and economical coil design.

**Keywords:** wireless power transmission; resonance; four-coil; relay coil

## 1. Introduction

Wireless power transmission (WPT) was first invented by Nikola Tesla more than 100 years ago and finally received attention for practical application in the early 2000s. Despite its advantage of being able to transmit power without wires, wireless power transmission has many limitations, such as a sharp decrease in power transmission efficiency depending on the distances and the positions of the transmitting and receiving coils. In 2007, researchers at the Massachusetts Institute of Technology developed a wireless power transmission technology method in the form of self-resonance. It can power a light bulb from a distance of 2 m. This wireless power transmission method using magnetic resonance makes it possible to exceed the limit of the present distance. In addition, research has been conducted on long-distance wireless power transmission using high-frequency microwaves. However, research and development in this area are limited owing to transmission efficiency and safety issues. The development of a resonant wireless power transmission method has garnered increased interest in wireless power transmission research. In particular, studies are being actively conducted with respect to the distance increasing and transmission efficiency increasing compared to the existing wireless power transmission method. The resonant wireless power transmission method thus has greater application potential.

Among some of the studies on the resonant wireless power transmission method are the following. A study was conducted on altering the material of the coil to improve transmission efficiency. Specifically, an optimization study was performed on the application of the coil material to the high-temperature superconducting coil [1]. Research

on omni-directional receivers was developed to reduce power transmission loss due to misalignment in wireless power transmission [2,3]. A study was undertaken to reduce the change in output due to the misalignment of electronic devices and electric vehicles using relay coils [4,5]. An analysis and design study strived to improve the Q-factor to increase the transmission distance in two-coil-type wireless power transmission [6]. Coil design and optimization studies of various structures and materials were conducted [7]. In one study, a transformer introduction model was designed for smallerization and increased efficiency in resonant wireless power transmission [8]. Research was conducted on the dipole shape of the transmitting and receiving coils to improve the wireless power transmission distance [9]. Moreover, a simulation study was conducted on improving the efficiency of wireless power transfer based on the winding material [10]. An experimental and simulation study was conducted on wireless power transfer for the double spiral coil shape [11], while another analyzed and optimized the misalignment of angles between coils [12]. Other studies analyzed the factors affecting wireless power transfer charging in electric transportation [13], while the characteristics of wireless power transmission by various mediums other than air, such as the sea, were investigated [14].

As shown in the above research, the shape, structure, and material of coils, including various media, and their optimization were assessed for application in various fields. The self-resonant wireless power transmission system consists of a source coil, transmission coil, and load coil. Each coil determines the inductance and capacitance values of the coil and capacitor to resonate with the system frequency. Therefore, to optimize the magnetic resonance method, not only the shape of the coil but also the relationship analysis of the coil's design value must be understood.

In this study, the coil was designed to have the resonance value of the system frequency. The transmission coil had the same resonance frequency; however, a model was designed through various combinations of inductance and capacitance, and the characteristics were analyzed through various experiments.

## 2. Composition and Operation of Four-Coil Magnetic Resonance

Unlike the induction-type wireless power transmission, self-resonant wireless power transmission consists of a total of four coils, as shown in Figure 1a, in a configuration in which a transmission coil and a reception coil are added. Each coil represents a structure in which the capacitor of the coil is connected in series so that the frequency of the input power becomes a resonance frequency. The resonant frequency is expressed as Equation (1).

$$f_r = \frac{1}{2\pi\sqrt{L_n C_n}} \tag{1}$$

where $f_r$ is the frequency of the system power source and becomes the resonance frequency of the coil. $L_n$ is the inductance determined by the magnetic resonance type wireless power transmission coil, and $C_n$ is the capacitance for obtaining the resonance frequency in each coil.

In the equivalent circuit diagram of Figure 1b, the current flowing through each coil is denoted as $I_1$, $I_2$, $I_3$, and $I_4$, and the input voltage is $V_S$. The determinant expressed in the Thévenin equivalent circuit is shown in Equation (2) [15,16].

$$\begin{bmatrix} V_s \\ 0 \\ 0 \\ 0 \end{bmatrix} = \begin{bmatrix} Z_1 & j\omega M_{12} & 0 & 0 \\ j\omega M_{12} & Z_2 & -j\omega M_{23} & 0 \\ 0 & -j\omega M_{23} & Z_3 & j\omega M_{34} \\ 0 & 0 & j\omega M_{34} & Z_4 \end{bmatrix} \begin{bmatrix} I_1 \\ I_2 \\ I_3 \\ I_4 \end{bmatrix} \tag{2}$$

where $Z_1$, $Z_2$, $Z_3$, and $Z_4$ represent the impedances of each coil, which are shown in Equation (3).

$$
\begin{aligned}
Z_1 &= R_s + R_1 + j\omega L_1 - j/\omega C_1 \\
Z_2 &= R_2 + j\omega L_2 - j/\omega C_2 \\
Z_3 &= R_3 + j\omega L_3 - j/\omega C_3 \\
Z_4 &= R_L + R_4 + j\omega L_4 - j/\omega C_4
\end{aligned}
\tag{3}
$$

In addition, $M_{12}$, $M_{23}$, and $M_{34}$ in Equation (2) represent the mutual inductance values between each coil, and they can be obtained using the following equation.

$$
M_{\mathrm{nm}} = k_{nm}\sqrt{L_n L_m}
\tag{4}
$$

Mutual inductances exist in coils that are not adjacent to each other, i.e., $M_{13}$, $M_{14}$, and $M_{24}$; however, they are not considered in wireless power transmission because the distance and structural effects are insignificant [15].

When analyzing the efficiency of the above system, it is advantageous to analyze it by considering it as a two-port network. Among the S-parameters, $S_{21}$ represents the ratio of the input signal to the output signal. Therefore, when the efficiency in the wireless power transmission system is indicated, it is denoted by the $S_{21}$ parameter. Its equation is the following:

$$
S_{21} = 2\frac{V_L}{V_S}\left(\frac{R_S}{R_L}\right)^{\frac{1}{2}}
\tag{5}
$$

Moreover, the $S_{21}$ parameters considering the mutual inductance in the $S_{21}$ parameters of Equation (5) are as follows:

$$
S_{21} = \frac{j2\omega^3 k_{12}k_{23}k_{34}L_2L_3\sqrt{L_1L_4R_SR_L}}{Z_1Z_1Z_1Z_1 + k_{12}^2L_1L_2Z_3Z_4\omega^2 + k_{23}^2L_2L_3Z_1Z_4\omega^2 + k_{34}^2L_3L_4Z_1Z_2\omega^2 + k_{12}^2k_{34}^2L_1L_2Z_3Z_4\omega^4}
\tag{6}
$$

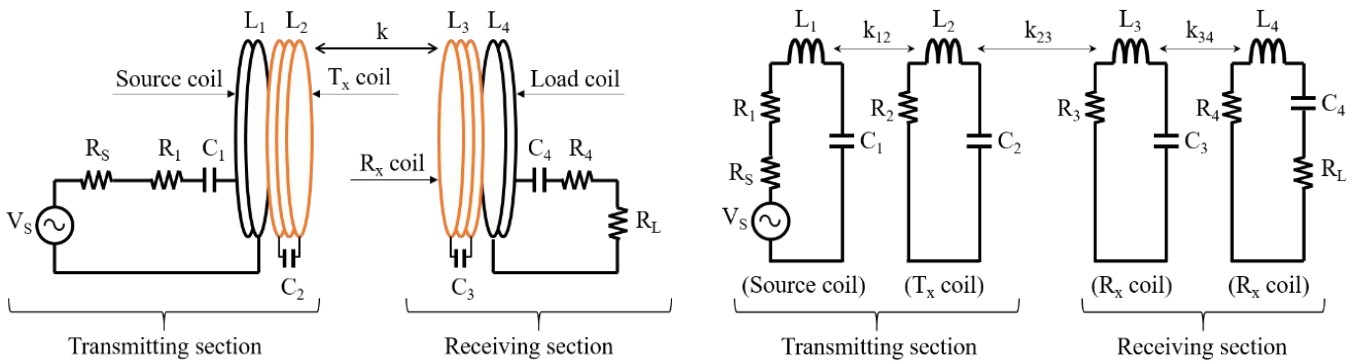

(**a**) Configuration diagram        (**b**) Equivalent circuit

**Figure 1.** A schematic diagram of a self-resonant wireless power transmission system.

### 3. Characteristics Analysis Method of Transmission Coil and Receiving Coil

The self-resonant wireless power transmission method is divided into two parts: a transmitting section and a receiving section. This format is the same as that of the magnetic induction-type wireless power transmission method. However, it differs from the magnetic induction-type wireless power transmission method in that it has relay coils, transmission coils, and receiving coils, which cause the designed coils to resonate at the power frequency. Self-resonant wireless power transmission enables longer-distance wireless power transmission compared to the magnetic induction method by means of the relay coil. Therefore, it is necessary to analyze the operating characteristics of the relay coil. To date, research has focused on analyzing the structural characteristics of the shape

of the coil. However, wireless power transmission requires electrical circuit analysis by transmitting power as it transmits its own changes. Therefore, circuit elemental analysis of the relay coil is required.

The relay coil is composed of a capacitor connected in series to the winding. It serves to transmit the winding magnetic field of the relay coil, and the capacitors are connected in series to match the resonance frequency of the relay coil. After configuring the winding of the relay coil, the capacitance is determined using Equation (1) based on the resonance frequency. To analyze the characteristics of the relay coil, the inductance of the winding is variously configured, as shown in Figure 2. The capacitance is determined to be resonant with the frequency of the input power source. As shown in Figure 2, the size of the inductance changes according to the number of windings, and the inductance increases as the number of windings increases; the resulting capacitance decreases. By analyzing the characteristics of magnetic resonant wireless power transmission for each coil's configuration, it is possible to derive an electrical circuit optimization model for the relay coil.

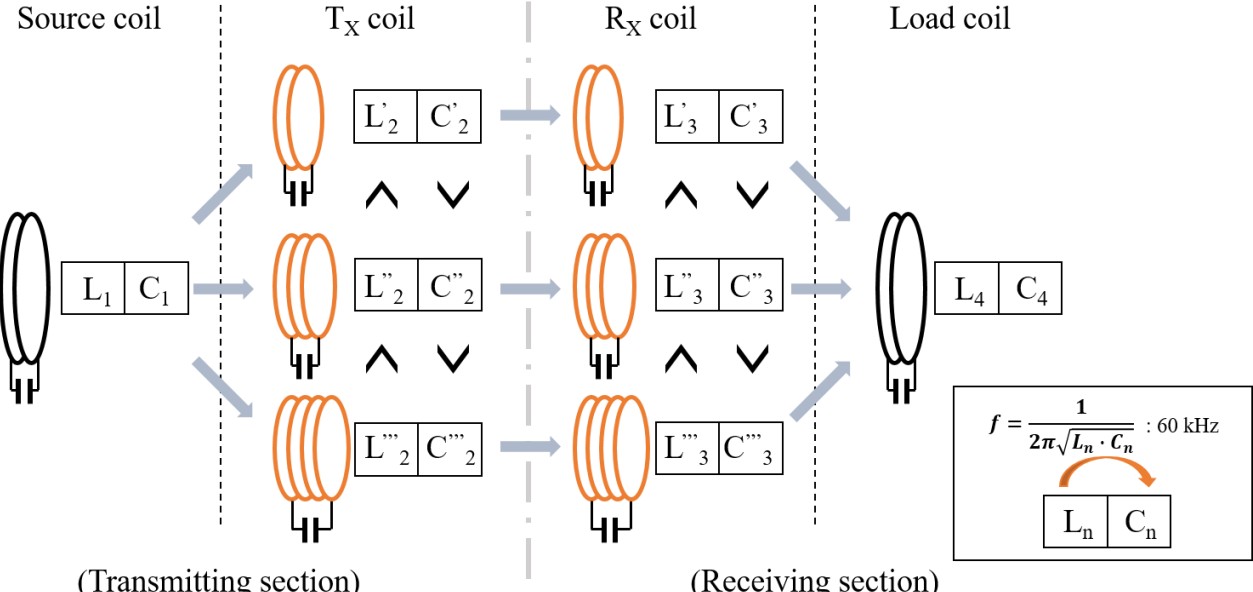

**Figure 2.** Overview of relay coil characterization method.

## 4. Experimental Setup

### 4.1. Sixty kHz, Design and Fabrication of Voltage Change Alternating Current Power Supplies

The frequency of the power used for wireless power transmission is different for each device, transmission method, and country. Depending on the transmission method, the magnetic induction method represents various frequencies, such as hundreds of kilohertz. The magnetic resonance method represents tens of kilohertz, and the microwave method represents a few kilohertz. Among various application fields, the wireless charging frequency of electric vehicles is 19–21 kHz, 55–65 kHz, and 79–90 kHz. Among them, the power is designed based on the frequency of 60 kHz most representatively used in the wireless charging of electric vehicles. The configuration diagram for the power design is as shown in Figure 3. To convert the inputted commercial power to a high frequency AC power of 60 kHz, the AC power of 60 Hz and up to 220 V is converted to DC through a full bridge rectifier. A transformer is used at the front end of the full bridge rectifier. The transformer is used to reduce the voltage at the output end of the transformer. It plays a key role in separating the circuit of the commercial power source and the circuit of the high-frequency generator. After charging the capacitor using the output of the full bridge rectifier, a high-frequency alternating current is generated through the half bridge inverter. For high-frequency power conversion, a metal–oxide–semiconductor field-effect

transistor—a power switch with a high switching operation—is used. The gate driver input signal for the switching operation is controlled by emitting a signal using NE555 and inputting it to the gate driver. A high-frequency AC power system of 60 kHz is constructed through the design and manufacture of this high-frequency generator power source.

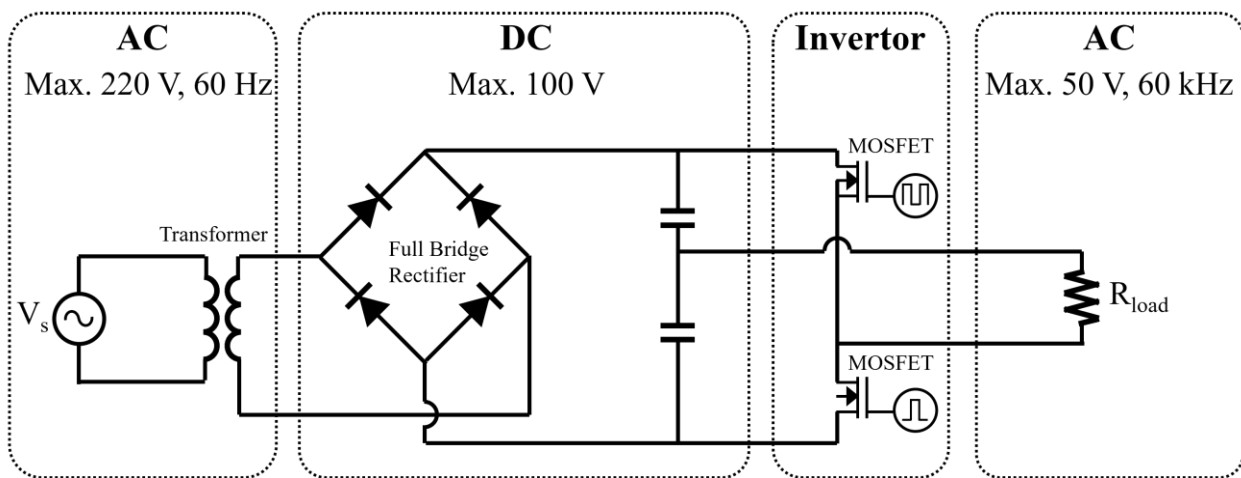

**Figure 3.** Diagram of high frequency AC generator.

Figure 4 is a photograph of a manufactured high-frequency AC generator. The frequency can be changed using the variable resistance of the gate driver circuit, and the fine voltage can be adjusted. The output waveform measured at the load resistance of the high-frequency AC generator is as shown in Figure 4b. The output voltage represents a square AC waveform that changes from approximately +10 V to −12 V. One cycle is approximately 17 µs, and the frequency is approximately 60 kHz.

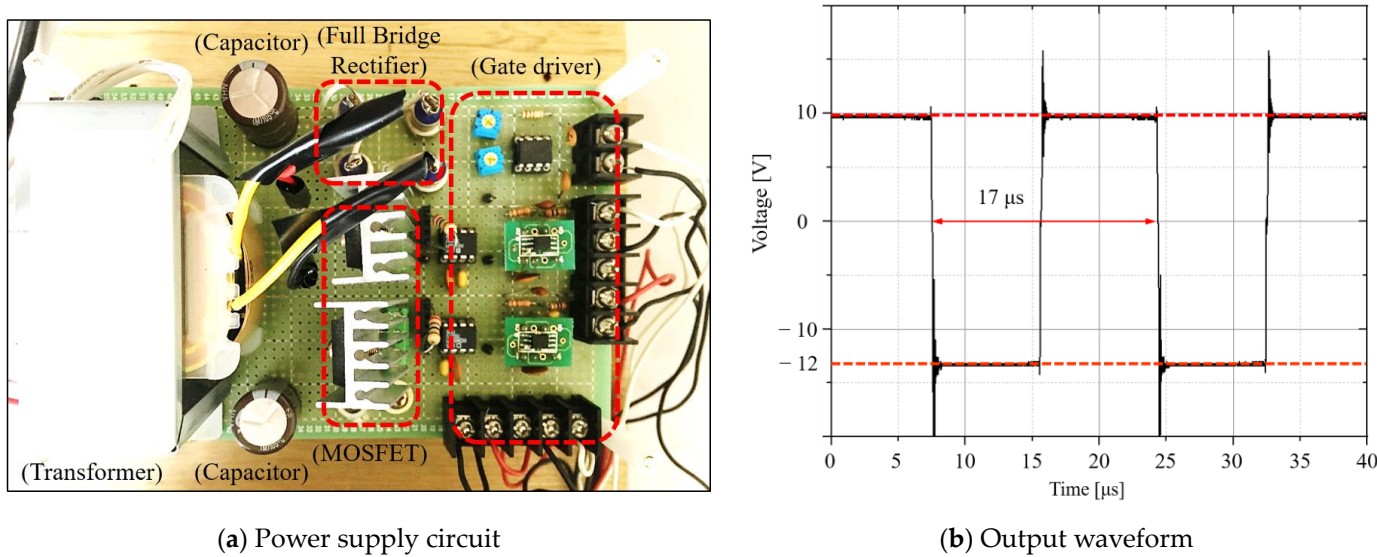

(**a**) Power supply circuit                    (**b**) Output waveform

**Figure 4.** Manufactured 10 V, 60 kHz output power supply.

### 4.2. The Design and Configuration of the Wireless Power Transmission Unit

The self-resonant wireless power transfer system consists of four coils: a source coil, Tx coil, Rx coil, and load coil. The relay coil (Rx coil, Tx coil) consists of a simple structure in which one inductor and one capacitor are connected in series. For each coil, capacitors are connected in series to the inductance of the winding so that the frequency of the input power becomes the resonance frequency. The configuration of the self-resonant wireless

power transfer system is shown in Figure 5a, and the configuration of the capacitor for the winding and resonance frequency of each coil is shown in Figure 5b. Because each coil becomes a resonance circuit for the input power, only the resistance component exists. In this study, since the frequency of the power input to the coil was 60 kHz, each coil was designed so that the resonance frequency was 60 kHz.

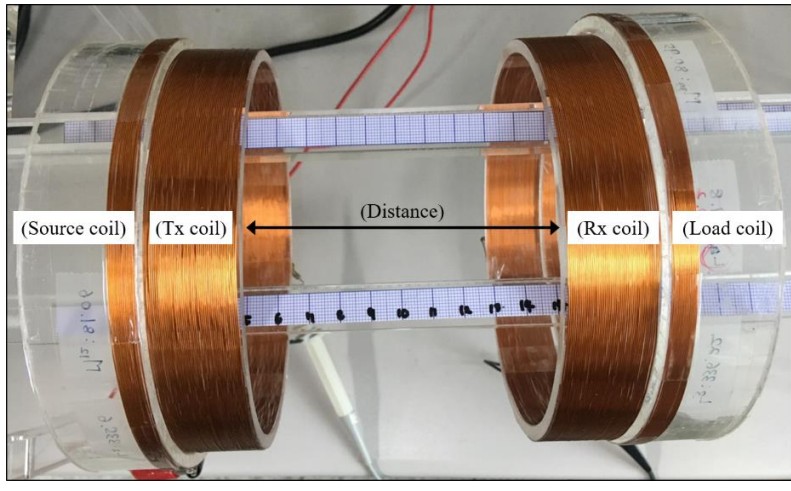

(**a**) Coil setup

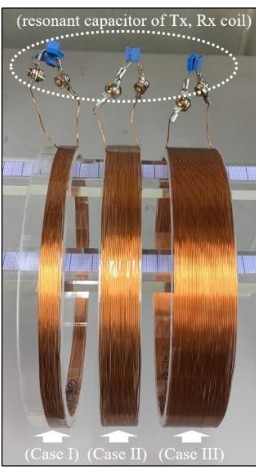

(**b**) Coil for each case

**Figure 5.** The configuration of the wireless power transmission unit.

In this study, three types of coils with different inductances were designed to analyze the relationship between the transmission efficiency of the inductance (turns) of the relay coil. Each design value is shown in Table 1.

**Table 1.** Design specifications for the wireless power transfer coil.

| Description | | | L [μH] | C [nF] | R [Ω] |
|---|---|---|---|---|---|
| Source coil (L1, C1) | | | 109.8 | 62 | - |
| Load coil (L4, C4) | | | 108.2 | 62 | - |
| Relay coil | Case I | Tx coil (L2, C2) | 95.6 | 73 | 1.38 |
| | | Rx coil (L3, C3) | 95.5 | 73 | 1.46 |
| | Case II | Tx coil (L2, C2) | 272.9 | 25 | 2.09 |
| | | Rx coil (L3, C3) | 282.3 | 25 | 2.14 |
| | Case III | Tx coil (L2, C2) | 634.3 | 11 | 2.41 |
| | | Rx coil (L3, C3) | 639.7 | 11 | 2.3 |

The inductance of the source coil and the load coil was set to 109.8 μH and 108.2 μH, respectively. That of Case I was designed to be 95.6 μH and 95.5 μH, that of Case II was designed to be 272.9 μH and 282.3 μH, and that of Case III was designed to be 634.3 μH and 639.7 μH. Based on the source coil and the load coil, in Case I, the ratio of the number of windings was approximately 1:1, in Case II, it was approximately 1:2, and in Case III, it was approximately 1:3. The internal resistance of the windings of Case I, Case II, and Case III are approximately 1.4 Ω, approximately 2.1 Ω, and approximately 2.4 Ω, respectively. To analyze the characteristics according to the inductance of the relay coil, the output voltage according to the distance of the coil was measured for each case. The distance was the interval between the Tx coil and the Rx coil, and the output voltage was measured while the interval was changed from 1 to 10 cm.

### 4.3. Experimental Results and Discussion

To experimentally analyze the characteristics according to the circuit values of the relay coils, Tx and Rx, a self-resonant wireless power transmission system was configured, as shown in Figure 6, based on the design of the power supply and that of the above transmission coil. The input power source was a square AC with a voltage of 10 V and a frequency of 60 kHz. The input power source could change the voltage and frequency. The input power source was capable of inputting up to 50 V to the source coil. The configuration of the transmission coil was composed of a total of four coils: a source coil, Tx coil, Rx coil, and load coil. Each coil was composed of the inductance and series capacitance of the winding so it could resonate with the input power frequency of 60 kHz. In the source coil and load coil, 50 $\Omega$ of system resistance $R_S$ and load resistance $R_L$ were respectively installed. The output voltage for each distance of the Tx coil and Rx coil was measured in $R_L$. A photograph of the wireless power transmission system configured to analyze the characteristics of the relay coil is shown in Figure 6b.

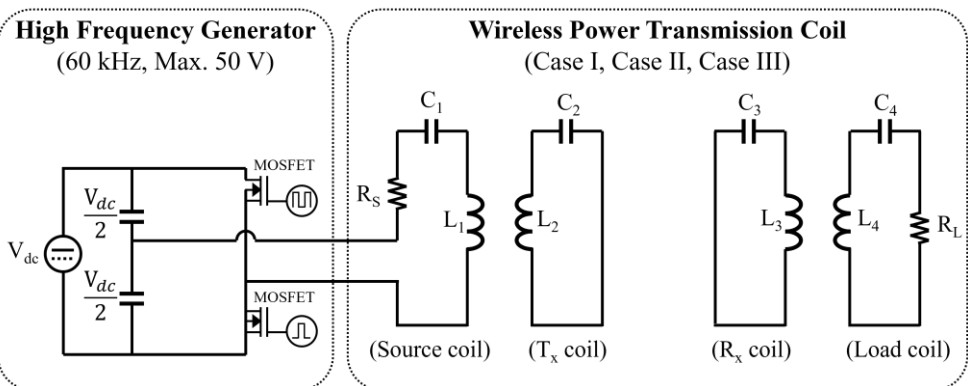

(**a**) The equivalent circuit model for the wireless power transfer system

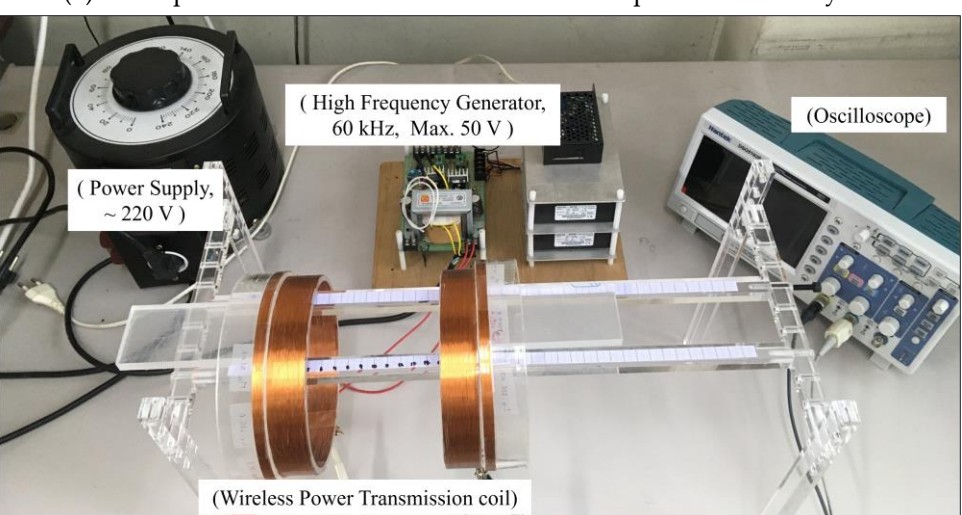

(**b**) Experiment setup

**Figure 6.** The configuration of the self-resonant wireless power transfer system.

The distance between the Tx coil and Rx coil for each case was measured three times, and the average value was calculated three times to compare and analyze the characteristics of the relay coil for each case. A graph of the average output voltage of Case I is shown in Figure 7. The maximum value of the average output voltage was measured at a distance of 3 cm and 5.49 V at 4 cm. In Case I, the maximum output was shown in the section from 3 cm to 4 cm. The measured voltage and average voltage for each distance in Case I are provided in Table 2.

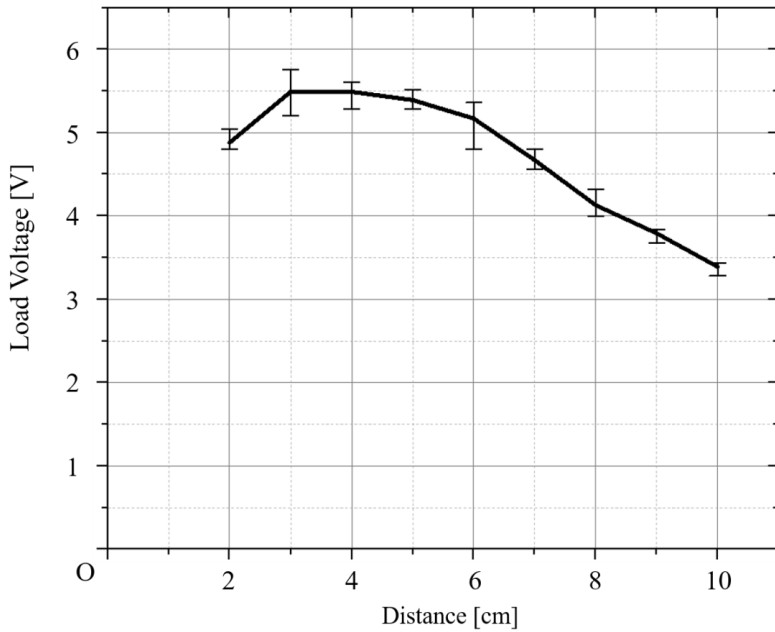

**Figure 7.** Graph of load voltage by distance in Case I.

A graph for the average output voltage for Case II is shown in Figure 8. The maximum value of the average output voltage is 5.97 V, and the distance between the coils is 4 cm. The measured voltage and average voltage for each distance of Case II are provided in Table 2.

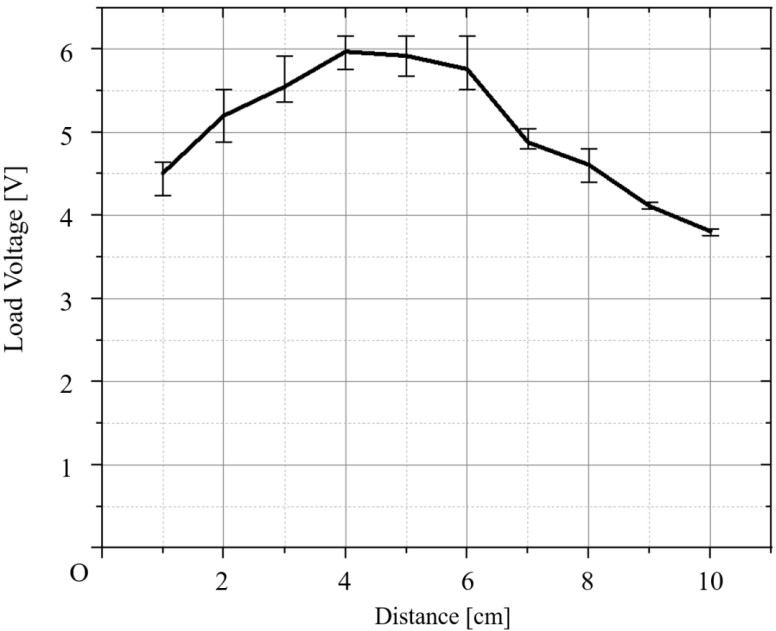

**Figure 8.** Graph of load voltage by distance in Case II.

A graph for the average output voltage for Case III is shown in Figure 9. The maximum value of the average output voltage is 5.95 V, and the distance between the coils is 3 cm. The measured voltage and average voltage for each distance of Case III are given in Table 2. The average values of the voltages measured by distance are shown at 3 cm and 4 cm in Case I, 4 cm in Case II, and 3 cm in Case III. For the measured voltages, the maximum voltage is 5.76 V at 3 cm, 6.16 V at 5 cm, and 6 V at 5 cm in Case III. The maximum measured value is 6.16 V in Case II. When comparing the output voltage graphs, it can be seen that the

output voltage of Case II and Case III shows similar values at each distance. And it was analyzed through experiments that the maximum voltage of Case I, Case II, and Case III was measured at similar positions, and this analysis become the characteristic of magnetic resonance wireless power transmission systems.

**Table 2.** Voltage measurements and mean value, efficiency.

| Distance | s/n | Case I | | | Case II | | | Case III | | |
|---|---|---|---|---|---|---|---|---|---|---|
| | | $V_{load}$ [V] | $V_{avr}$ [V] | Efficiency ($S_{21}$) | $V_{load}$ [V] | $V_{avr}$ [V] | Efficiency ($S_{21}$) | $V_{load}$ [V] | $V_{avr}$ [V] | Efficiency ($S_{21}$) |
| 1 cm | 1 | - | - | - | 4.64 | 4.51 | 0.71 | 4.64 | 4.72 | 0.74 |
| | 2 | - | | | 4.64 | | | 4.64 | | |
| | 3 | - | | | 4.24 | | | 4.88 | | |
| 2 cm | 1 | 4.8 | 4.88 | 0.77 | 5.52 | 5.20 | 0.82 | 5.6 | 5.39 | 0.85 |
| | 2 | 5.04 | | | 5.2 | | | 5.2 | | |
| | 3 | 4.8 | | | 4.88 | | | 5.36 | | |
| 3 cm | 1 | 5.2 | 5.49 | 0.86 | 5.36 | 5.55 | 0.87 | 6 | 5.95 | 0.93 |
| | 2 | 5.76 | | | 5.36 | | | 5.92 | | |
| | 3 | 5.52 | | | 5.92 | | | 5.92 | | |
| 4 cm | 1 | 5.6 | 5.49 | 0.86 | 6 | 5.97 | 0.94 | 5.76 | 5.87 | 0.92 |
| | 2 | 5.28 | | | 5.76 | | | 5.92 | | |
| | 3 | 5.6 | | | 6.16 | | | 5.92 | | |
| 5 cm | 1 | 5.36 | 5.39 | 0.85 | 6.16 | 5.92 | 0.93 | 5.76 | 5.92 | 0.93 |
| | 2 | 5.52 | | | 5.68 | | | 6.0 | | |
| | 3 | 5.28 | | | 5.92 | | | 6.0 | | |
| 6 cm | 1 | 4.8 | 5.17 | 0.81 | 5.52 | 5.76 | 0.90 | 5.68 | 5.49 | 0.86 |
| | 2 | 5.36 | | | 6.16 | | | 5.6 | | |
| | 3 | 5.36 | | | 5.6 | | | 5.2 | | |
| 7 cm | 1 | 4.56 | 4.67 | 0.73 | 4.8 | 4.88 | 0.77 | 4.88 | 5.04 | 0.79 |
| | 2 | 4.8 | | | 4.8 | | | 5.2 | | |
| | 3 | 4.64 | | | 5.04 | | | 5.04 | | |
| 8 cm | 1 | 4.32 | 4.13 | 0.65 | 4.4 | 4.61 | 0.72 | 4.56 | 4.64 | 0.73 |
| | 2 | 4.08 | | | 4.8 | | | 4.56 | | |
| | 3 | 4 | | | 4.64 | | | 4.8 | | |
| 9 cm | 1 | 3.84 | 3.79 | 0.59 | 4.08 | 4.11 | 0.65 | 4.32 | 4.35 | 0.68 |
| | 2 | 3.84 | | | 4.16 | | | 4.32 | | |
| | 3 | 3.68 | | | 4.08 | | | 4.4 | | |
| 10 cm | 1 | 3.44 | 3.39 | 0.53 | 3.84 | 3.81 | 0.60 | 3.92 | 3.84 | 0.60 |
| | 2 | 3.44 | | | 3.84 | | | 3.76 | | |
| | 3 | 3.28 | | | 3.76 | | | 3.84 | | |

To show the efficiency of power transmission for each case, Equation (5) is used and compared with the $S_{21}$ parameter. Its graph is shown in Figure 10. The efficiency of the maximum power transmission in the $S_{21}$ parameter is shown at the 4 cm distance between the coils of Case II. The rate of increase in efficiency between each case does not show a constant increase for each distance; thus, it is difficult to simply compare them. Therefore,

the rate of increase between the cases is compared at 3 cm and 4 cm in the distance between the coils, which represent the maximum average voltage for each case. In Case I, the increase rate of Case II shows an increase of approximately 1% at a distance of 3 cm and approximately 9% at 4 cm. In Case II, the increase rate for Case III is approximately 7% at 3 cm, and the decrease rate is approximately 2% at 4 cm. As shown in Figure 10, the power transmission efficiency increases when the inductance (turns) of the relay coil is increased. However, when comparing Case II and Case III, it is observed that no significant difference exists in transmission efficiency. Accordingly, the power transmission efficiency increases as the inductance of the relay coil increases. However, with the increase in inductance, as in the comparison of Case II and Case III, a saturated section occurs that no longer increases with the improvement in power transmission efficiency. When comparing the winding number, it is observed that the winding number ratio of the source coil, the load coil, and the relay coil increases the efficiency of power transmission at the winding ratio of 1:2 rather than that of 1:1.

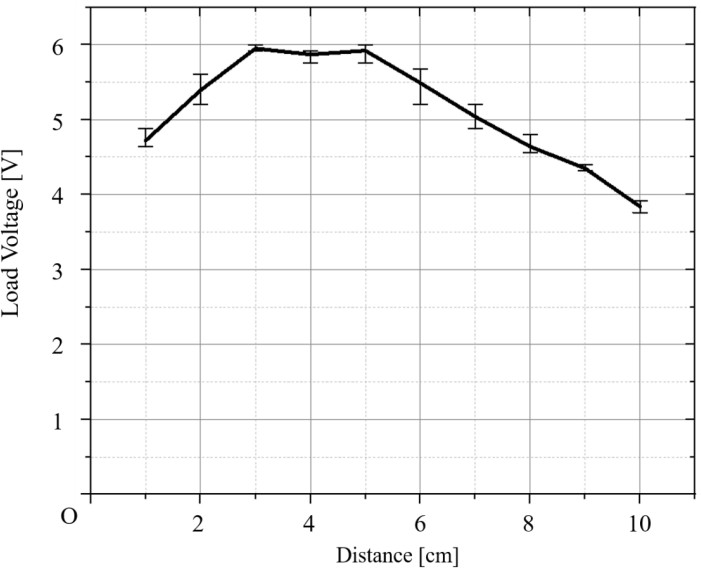

**Figure 9.** Graph of load voltage by distance in Case III.

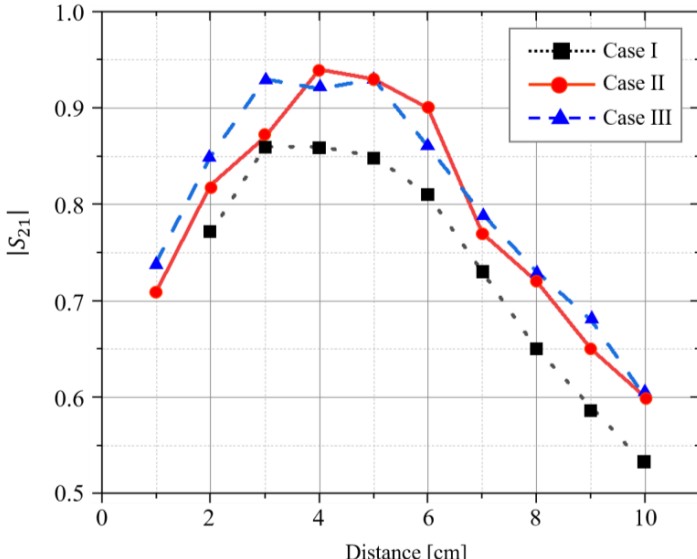

**Figure 10.** Graph of S21 parameter in Cases I, II, and III.

In the results of this experiment, the optimal winding ratio of the source coil (load coil) to the relay coil (Rx coil) is confirmed as 1:2. Therefore, it can be confirmed that the inductance ratio of the source coil (load coil) and the relay coil (Tx coil) of the optimal economical relay coil design is 1:3, and the winding ratio is 1:2.

## 5. Conclusions

This study was conducted to analyze power transmission efficiency according to the relay coil inductance of the self-resonant wireless power transmission system. To this end, the optimal design of the coil was established by assessing the magnitude of the $S_{21}$ parameter. The coil was designed so that the inductance of the relay coil was approximately one, three, and six times the inductance of the source coil (load coil), and the winding ratio with the source coil (load coil) was 1:1, 1:2, and 1:3. The distance representing the maximum efficiency was between 3 and 5 cm. As the inductance value increased by one to three times, the efficiency increased by approximately 10%. However, as the inductance increased from three to six times, the change in efficiency decreased. Based on the above experimental results, it is possible to analyze the operating characteristics according to the circuit analysis of the relay coil as follows.

- As the inductance of the coil increases, the magnitude of the $S_{21}$ parameter increases. This finding indicates that the efficiency of wireless power transmission increases.
- It is confirmed that as the inductance increases, the amount of the efficiency increase gradually decreases at some point.
- In the magnetic resonance wireless power transmission system, each coil has an air-core transformer structure, and an increase in inductance (turns) causes a voltage increase. Therefore, this reduces losses and improves power transmission efficiency.
- As the number of turns of the relay coil increases, the induced voltage increases, and the current flowing through the coil increases accordingly.
- As the inductance (turns) of the relay coil increases, the increase in internal resistance causes a limit to the increase in efficiency.

As shown in the above results, when the relay coil has a larger inductance value than the source coil (load coil), the power transmission efficiency increases. Nevertheless, the experimental results show that the increase in efficiency decreases at some point. Therefore, for maximum power transmission efficiency, the ratio of the inductance of the source coil to the relay coil is believed to require a winding design between three and six times.

In the future, based on the results of this paper, we plan to conduct research on methods to focus magnetic flux, such as analyzing characteristics according to the radius of the relay coil. And we will analyze the effect of the internal resistance of the winding on wireless power transmission. Therefore, it is possible to increase distance, and increase without additional devices, and it is believed that it can be applied to various fields.

**Author Contributions:** Conceptualization, M.-W.H. and K.-C.K.; Methodology, K.-C.K.; Formal analysis, Y.-M.K.; Investigation, M.-W.H.; Data curation, M.-W.H. and Y.-M.K.; Writing—original draft, M.-W.H. and Y.-M.K.; Writing—review & editing, K.-C.K.; Visualization, M.-W.H. and Y.-M.K.; Supervision, K.-C.K. All authors have read and agreed to the published version of the manuscript.

**Funding:** This research received no external funding.

**Data Availability Statement:** Data are contained within the article.

**Conflicts of Interest:** The authors declare no conflict of interest.

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
