# Peer review of "Optimal Design of Relay Coil Inductance to Improve Transmission Efficiency of Four-Coil Magnetic Resonance Wireless Power Transmission Systems"

_electronics, doi:10.3390/electronics13071261_

Round 1

Reviewer 1 Report

Comments and Suggestions for Authors

The manuscript “Optimal Design of Relay Coil Inductance to Improve Transmission Efficiency of Four-Coil Magnetic Resonance Wireless Power Transmission Systems” focuses on the optimization of relay coil inductance to enhance transmission efficiency in magnetic resonance wireless power transmission systems. The study presents a comprehensive analytical framework and experimental validation aimed at increasing the transmission distance and improving transmission efficiency. In my assessment, the manuscript is methodologically sound and provides valuable insights for the optimization of wireless power transmission systems. I would like to recommend the publication of this manuscript in Electronics pending some proper revisions / responses.

1. The authors mentioned in the manuscript that the ratio of the inductance of the source coil to the relay coil is believed to require a winding design between three and six times for maximum power transmission efficiency. Please analyze the reasons and the theoretical basis for optimizing the wireless power transmission systems.

2. The authors discussed in the manuscript how to optimize the wireless power transmission system by optimizing the inductance value of the relay coil. Is there any other method to improve the transmission efficiency and transmission distance of wireless power transmission systems? Please briefly discuss and analyze it.

3. The authors mentioned that the optimal design of the relay coil is believed to be the most efficient and economical coil design. In fact, recently, efficient and stable wireless power transfer can be realized assisted by the non-Hermitian physics with purely real eigenvalues:

(1) Robustness of Wireless Power Transfer Systems with Parity-Time Symmetry and Asymmetry, https://www.mdpi.com/1996-1073/16/12/4605

(2) Rotation manipulation of high-order PT-symmetry for robust wireless power transfer, https://www.sciencedirect.com/science/article/pii/S2667325823003151

(3) Level pinning of anti-PT symmetric circuits for efficient wireless power transfer, https://academic.oup.com/nsr/article/11/1/nwad172/7198125

These highly related works should draw the authors’ attention.

Comments on the Quality of English Language

Minor editing of English language required

Author Response

Thanks to the reviewer, we were able to fix and improve our manuscript. We revised the manuscript to follow the review’s instructions. Below is our answer to the inquiries that the reviewer pointed.

Point 1: The authors mentioned in the manuscript that the ratio of the inductance of the source coil to the relay coil is believed to require a winding design between three and six times for maximum power transmission efficiency. Please analyze the reasons and the theoretical basis for optimizing the wireless power transmission systems.

Response 1: The four-coil resonance method consists of source coil, Tx coil, Rx coil, and load coil. The energy is transferred from each coil to the next, and maximum energy transfer occurs at the resonance frequency.

In the four-coil method, the structure between each coil is the same as that of the air core transformer. In power transmission, as the voltage is increased, the loss decreases, resulting in an increase in transmission efficiency. As such, it can be seen that the efficiency increases as the winding ratio of the source coil and the relay coil doubles in the proposed model of this paper.

However, it was confirmed through an experiment that the increase rate decreased as the winding ratio increased three times. It was analyzed to be the effect of limiting the current flowing through the winding as the parasitic resistance increases as the winding ratio increases.

Point 2: The authors discussed in the manuscript how to optimize the wireless power transmission system by optimizing the inductance value of the relay coil. Is there any other method to improve the transmission efficiency and transmission distance of wireless power transmission systems? Please briefly discuss and analyze it.

Response 2: As explained in Response 1, each coil has a structure of an air core transformer. The boosting ratio varies depending on the inductance (winding ratio) of the relay coil and affects the energy transfer efficiency accordingly. And it shows an improvement in energy transfer efficiency for the same distance.

Point 3: The authors mentioned that the optimal design of the relay coil is believed to be the most efficient and economical coil design. In fact, recently, efficient and stable wireless power transfer can be realized assisted by the non-Hermitian physics with purely real eigenvalues:

(1) Robustness of Wireless Power Transfer Systems with Parity-Time Symmetry and Asymmetry, https://www.mdpi.com/1996-1073/16/12/4605

(2) Rotation manipulation of high-order PT-symmetry for robust wireless power transfer, https://www.sciencedirect.com/science/article/pii/S2667325823003151

(3) Level pinning of anti-PT symmetric circuits for efficient wireless power transfer, https://academic.oup.com/nsr/article/11/1/nwad172/7198125

These highly related works should draw the authors’ attention.

Response 3: Thank you for your careful guidance. I will refer to your suggestions for further efficiency improvement research and optimization research in the future.

Reviewer 2 Report

Comments and Suggestions for Authors

This manuscript presents the optimization of relay coil inductance for magnetic resonance wireless power transfer through analyzing the experimental results of the distance effect on the power transmission efficiency. The contents are easy to follow, and the presentation is good. Below are some comments for improving the manuscript:

1.      English has to be checked carefully – e.g., Lines 60-61: Another studied analyzed the factors affecting…

2.      Reference for Equation (2) has to be included so that readers can refer to the detailed conditions that make Eq. (2) hold and applicable.

3.      Line 157 (Fig 4) & Line 190 say 10V (variable) was adopted, while in Fig 6 the high freq generator provided ~50V. Please confirm.

4.      Figures 7 to 9 should be combined into a single figure, just like Fig 10.

5.      Since k12 and k34 are apparently zero in this work, the authors have to clearly state that point and discuss more in-depth effects of k12 & k34 so that the points in Conclusions can provide a clearer picture for this work.

6.      Statements for future work and/or suggestive directions will be necessary for enhancing the novelty of this work.

7.      Missing year infor of ref [8] – please duly check the list of references.

Comments on the Quality of English Language

Moderate editing of English language required

Author Response

Thanks to the reviewer, we were able to fix and improve our manuscript. We revised the manuscript to follow the review’s instructions. Below is our answer to the inquiries that the reviewer pointed.

Point 1: English has to be checked carefully – e.g., Lines 60-61: Another studied analyzed the factors affecting.

Response 1: The overall contents were carefully reviewed and revised.

Point 2: Reference for Equation (2) has to be included so that readers can refer to the detailed conditions that make Eq. (2) hold and applicable.

Response 2: A reference for Equation (2) has been added to the paper.

Point 3: Line 157 (Fig 4) & Line 190 say 10V (variable) was adopted, while in Fig 6 the high freq generator provided ~50V. Please confirm.

Response 3: The meaning of 50V in figure 6 means the maximum supply-able voltage. As your comments, in Figure 6, ~50V was modified to Max. 50.

Point 4: Figures 7 to 9 should be combined into a single figure, just like Fig 10.

Response 4: In order to compare the magnitudes of the output voltages in Case I, II, and III, the reviewer is correct. However, the graph shown in Figure 7 to 9 is a graph that measures three times for each distance and represents the average value, and I think it is meaningful to check individual values through the error bar.

Point 5: Since k12 and k34 are apparently zero in this work, the authors have to clearly state that point and discuss more in-depth effects of k12 & k34 so that the points in Conclusions can provide a clearer picture for this work.

Response 5: k12 and k34 are not zero. If they are zero, there is no mutual inductance, and no energy transfer will take place.

In four-coil resonant wireless power transmission, mutual inductance exists between each coil, and energy is transferred sequentially.

As mentioned in the paper, values such as k13, k14, and k24 exist, but are very small and negligible. Accordingly, it was considered that mutual inductance also did not exist.

Point 6: Statements for future work and/or suggestive directions will be necessary for enhancing the novelty of this work.

Response 6: In the conclusion, based on this paper, we added the contents of future research.

Point 7: Missing year infor of ref [8] – please duly check the list of references.

Response 7: I checked the reference as a whole and correctly rewritten it.
